# Fabrication of arbitrary three-dimensional suspended hollow microstructures in transparent fused silica glass

Frederik Kotz[1], Patrick Risch[1], Karl Arnold[1], Semih Sevim[2], Josep Puigmartí-Luis[2], Alexander Quick [3], Michael Thiel[3], Andrei Hrynevich[4], Paul D. Dalton[4], Dorothea Helmer[1] & Bastian E. Rapp [1]

Fused silica glass is the preferred material for applications which require long-term chemical and mechanical stability as well as excellent optical properties. The manufacturing of complex hollow microstructures within transparent fused silica glass is of particular interest for, among others, the miniaturization of chemical synthesis towards more versatile, configurable and environmentally friendly flow-through chemistry as well as high-quality optical wave-guides or capillaries. However, microstructuring of such complex three-dimensional structures in glass has proven evasive due to its high thermal and chemical stability as well as mechanical hardness. Here we present an approach for the generation of hollow microstructures in fused silica glass with high precision and freedom of three-dimensional designs. The process combines the concept of sacrificial template replication with a room-temperature molding process for fused silica glass. The fabricated glass chips are versatile tools for, among other, the advance of miniaturization in chemical synthesis on chip.

[1] NeptunLab, Laboratory of Process Technology, Department of Microsystems Engineering (IMTEK), University of Freiburg, Georges-Köhler-Allee 103, Freiburg 79110, Germany. [2] Institute for Chemical & Bioengineering, Department of Chemistry & Applied Biosciences, ETH Zurich, 8083 Zurich, Switzerland. [3] Nanoscribe GmbH, Hermann-von-Helmholtz-Platz 1, 76344 Eggenstein-Leopoldshafen, Germany. [4] Department for Functional Materials in Medicine and Dentistry and Bavarian Polymer Institute, University Würzburg, Pleicherwall 2, 97070 Würzburg, Germany. Correspondence and requests for materials should be addressed to B.E.R. (email: Bastian.Rapp@imtek.de)

Microstructures in fused silica glass are usually fabricated by wet chemical or dry etching processes[1]. More complex structures can be fabricated using precision glass molding, sol-gel replication or powder blasting[2–4]. However, all these techniques are only capable of fabricating open, two-dimensional channel structures, which require bonding with a planar substrate to fabricate simple suspended hollow microstructures (e.g., microfluidic channels). Creating freeform hollow structures inside fused silica glass is difficult and has, until now, only been shown using femtosecond laser writing with consecutive etching of the irradiated areas with aggressive chemicals such as hydrofluoric acid (HF)[5,6]. However, for long channel structures with few inlets the etching is inhomogeneous and results in tapered channel structures with significantly wider dimensions towards the channel inlets[7]. Additionally, channel lengths are limited by the etching process, since HF etching shows a decrease in etching speed over channel length and debris can quickly block the channels[8]. To overcome the problem of varying channel diameters, different techniques such as wobbling or drawing have been established[9,10]. However, these techniques can only be employed for simple channel geometries. To overcome the need for aggressive etching solutions, femtosecond laser writing by liquid-assisted ablation was developed[11,12]. This technique generates components with significant surface roughness, which require post-treatment to generate surfaces of optical quality[5,13]. A method for three-dimensional structuring of high-silica glasses with an $SiO_2$ content of 95–97% via femtosecond laser writing has been previously reported, using a porous glass similar to VYCOR[14]. The structures produced, however, display a coarse wall structure and non-uniform channel cross-sections. A comparison of relevant techniques to structure fused silica glass can be found in Table 1.

As of today there is no method for generating truly arbitrary three-dimensional hollow structures of centimeter lengths and few micrometers diameter in bulk fused silica glass. However, many applications such as, e.g., microfluidics, flow-through synthesis, photonics or waveguiding applications in optics, and photonics require methods for creating freeform hollow structures in fused silica with smooth surfaces. These structures are also highly sought for flow-through on-chip chemical synthesis, a field which has recently gained significant attention[15,16]. Miniaturization of chemical reactions promises significantly reduced reactant consumption, more stable reaction conditions and new reaction pathways such as ultrafast mixing or kinetic reaction control, which are inaccessible in standard batch and flask chemistry[17,18]. Recently, the use of additive manufacturing for the fabrication of configurable, low-volume synthesis systems has gained significant attention[19]. However, suitably high resolution structures can, until now, only be manufactured in polymers, which in turn limits the choice of solvents, temperature and pressure. In many applications the established chemistries (optimized for glassware) cannot be directly translated to polymers and significantly lower reaction yields and efficiencies have been achieved. In a recent contribution to Science, Kitson et al., stated that for polymer-based reactionware suitable for on-demand pharmaceutical synthesis, a translation process from glassware to polymers would be necessary[20]. As glass is the material of choice to withstand the harsh reaction conditions as well as to enable the on-line analysis of reactions through spectroscopy, novel approaches to manufacturing of intricate three-dimensional glass structures are highly sought.

We have recently developed a method for structuring fused silica components at room temperature[21–23]. In this process, a nanocomposite consisting of a high amount of fused silica nanoparticles in an organic binder matrix is polymerized at room temperature and consecutively sintered to full-density, transparent fused silica glass. We have shown that using stereolithography three-dimensional fused silica glass structures can be fabricated[22]. Fabrication of suspended three-dimensional hollow microstructures by 3D printing, however, remains intrinsically difficult since entrapped uncured material inside the microvoids is difficult to remove and is partially cured during the printing process, thereby blocking the microstructures. Here we demonstrate that a combination of the casting of these nanocomposites and sacrificial template replication (STR), a concept known from polymer and ceramic processing, is able to produce complex, suspended hollow microstructures in fused silica glass. In sacrificial template replication, a template structure or immiscible phase is introduced into a material and consecutively removed by dissolving, etching or burning to produce a desired hollow structure[24–28]. Nanochannels in glass can be produced by coating of electrospun nanofibers with silicon dioxide and consecutive calcination, but have so far not been shown for fused silica and are restricted to single electrospun fibers and therefore limited to very simple designs[29]. Such sacrificial template techniques have previously been used in combination with metallic glasses and bioglasses. However, the structures produced are restricted to open structures with lamellae or fibrous/porous surface structures[30,31]. By

**Table 1 Comparison of relevant methods for structuring fused silica glass**

| Method | 3D capability | 3D microvoids[a] | Resolution | Surface quality | Literature |
|---|---|---|---|---|---|
| *Etching* | | | | | |
| Wet chemical etching | − | − | ~1 μm | ~1–10 nm (Ra) | 37,38 |
| Dry etching | − | − | <1 μm | 0.5 (rms)–2 nm (Ra) | 39,40 |
| *Mechanical* | | | | | |
| Powder blasting | − | − | >10 μm | 0.1–10 μm (Ra) | 41 |
| *Laser-assisted* | | | | | |
| Laser-assisted etching | + | + | 1–2 μm | 0.1–0.2 μm (rms) | 42–44 |
| Backside etching | − | − | 2 μm | 0.05–0.5 μm | 45 |
| *Replication* | | | | | |
| Sol-gel | − | − | <1 μm | n.a. | 3 |
| Nanocomposites | − | − | <1 μm | 2 nm (rms) | 21,23 |
| Precision glass molding | − | − | ~1 μm | 2 nm | 2,46 |
| *Additive* | | | | | |
| Stereolithography nanocomposites | ++ | − | 60 μm | 2 nm (rms) | 22 |
| Sol-gel | ++ | − | 200 μm | n.a. | 47 |
| Stop flow lithography | − | − | 10 μm | 6 nm (rms) | 48 |

[a]Suitability to create microvoids with a size of 1–100 μm

combining room temperature glass structuring and sacrificial template replication, it is possible, to generate nearly arbitrarily-shaped freeform three-dimensional channels and hollow structures within fused silica glass.

## Results

**Sacrificial template replication**. We show template molding using nylon threads, poly(ethylene glycol diacrylate) (PEGDA) scaffolds, poly(ε-caprolactone) (PCL) microfiber meshes produced by melt electrowriting, and complex polymeric microstructures fabricated by direct laser writing. All scaffolds were immersed in the nanocomposite[21] and consecutively processed to decompose and evaporate the polymeric residue and to give fused silica glass microstructures. This is an advantage over many sacrificial template replication process like the fugitive ink technology where the template has to be liquefied and washed out of the bulk material[32]. As the templates are removed in the gas phase, there is no material redeposition or channel blocking by incomplete removal. In addition, diffusion limitations for the removal of the template do not apply. As the templates are removed in the gas phase, there is no material redeposition or

channel blocking by incomplete removal. Also diffusion limitations, which usually restrict the length and dimensions of etched structures, do not apply. The STR process in fused silica glass is shown in Fig. 1a. First the polymeric template is embedded in the nanocomposite. The nanocomposite is then polymerized using light exposure. Thermal debinding of the polymeric binder and the template is done at 600 °C and ambient pressure. The structures are consecutively sintered at a pressure of $5 \times 10^{-2}$ mbar at 1300 °C (see Supplementary Table 1 for the optimized protocol for thermal debinding and sintering). During the sintering process the parts shrink isotropically in dependence of the solid loading. Here we used a nanocomposite with a solid loading of 40 vol% resulting in a linear shrinkage of 26.3%. For example the length of the upper channel of the mixer in Fig. 2e showed the expected linear shrinkage of 26.28% from 2121.95 μm to 1564.23 μm. Further information on the shrinkage calculation can be found in the supplementary information.

We have recently demonstrated that the sintered fused silica glass parts show the same high optical transparency in the UV, visible and infrared region as well as the same mechanical, chemical and thermal stability. We further demonstrated that the

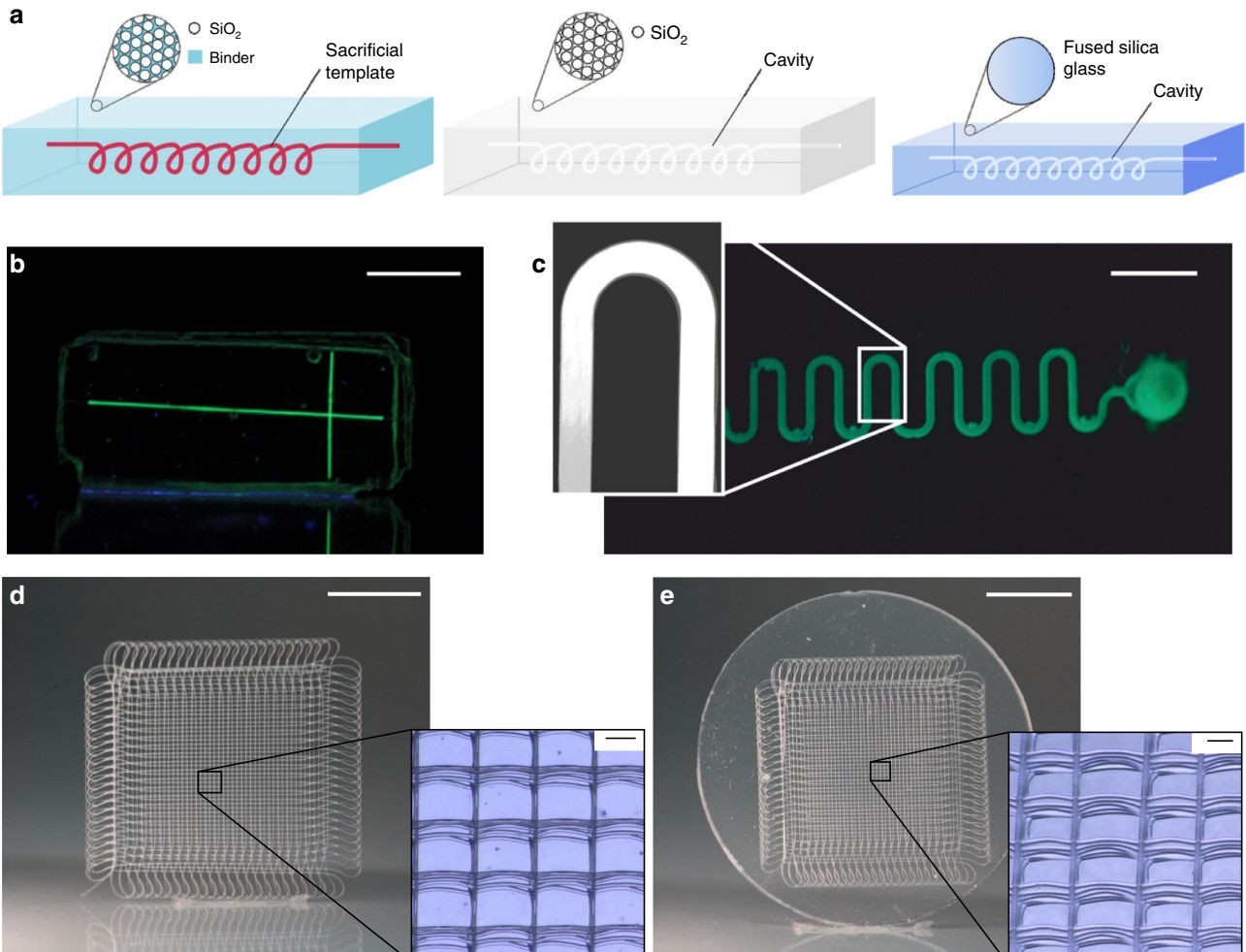

**Fig. 1** Fabrication of suspended hollow microstructures in fused silica glass. **a** Polymeric filaments are embedded in an amorphous silica nanocomposite. The polymerized nanocomposite is turned into fused silica glass via thermal debinding and sintering. The polymeric template is removed during the thermal debinding process and leaves the according hollow cavity. **b** Microfluidic fused silica chip fabricated by embedding a nylon thread (scale bar: 9 mm). **c** Microfluidic meander fabricated by embedding polymerized PEGDA structured by microlithography (scale bar: 11 mm). **d** A mesh structure made from poly(ε-caprolactone) using melt electrowriting (scale bar: 5 mm). The inset shows the microscopy image of the mesh with a fiber diameter of 25.0 μm (scale bar: 100 μm). **e** Inverse hollow mesh structure in fused silica glass (scale bar: 4.5 mm). Inset shows the microcavities with a width of around 18.4 μm (scale bar: 100 μm)

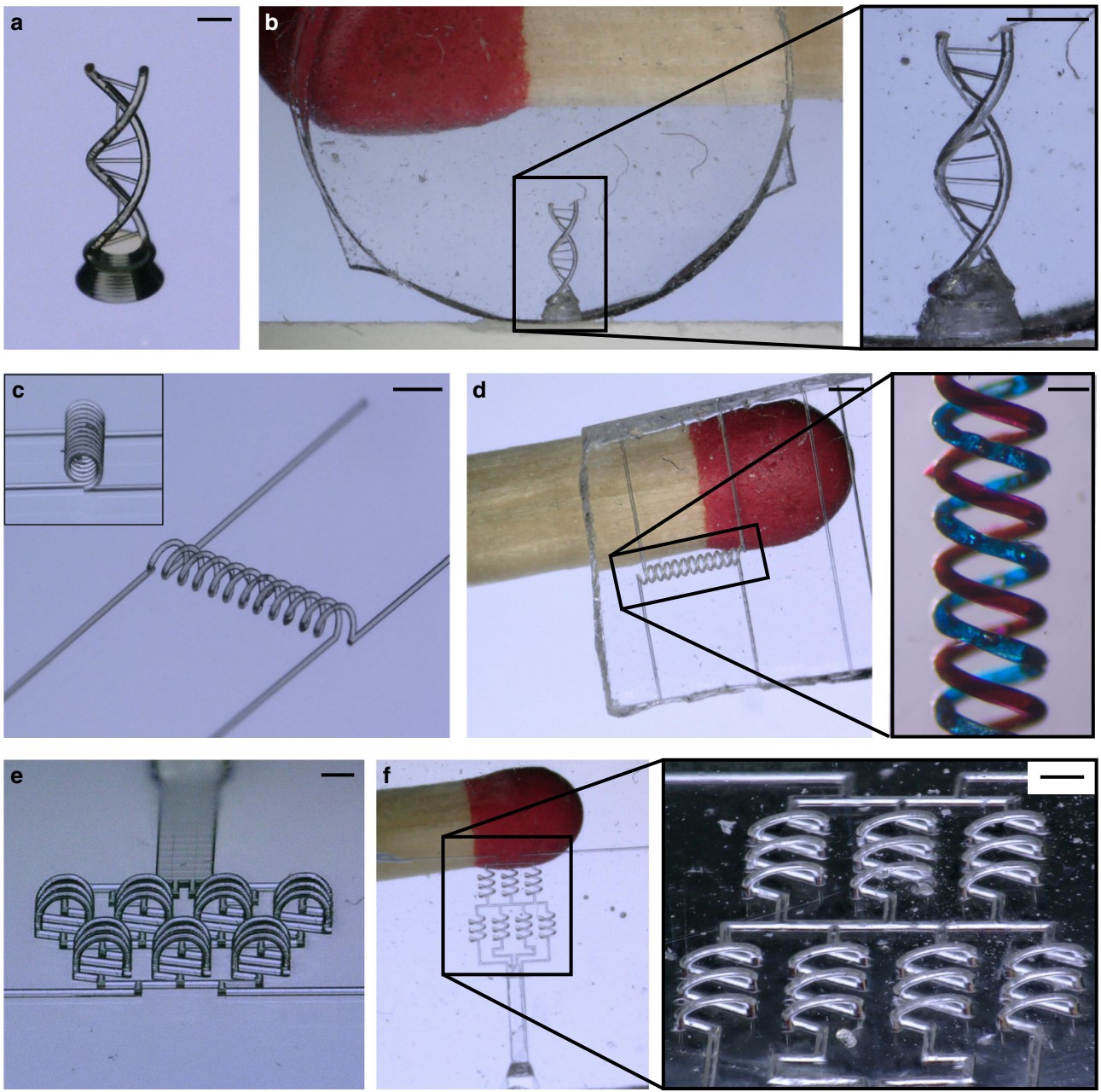

**Fig. 2** STR using templates produced by direct laser writing. **a** Polymeric DNA double-helix (scale: 500 μm). **b** Inverse structure in fused silica glass (scale: 400 μm). The smallest channel size is 20 μm. **c** Intertwined spirals (scale: 900 μm). **d** Resulting intertwined microfluidic spiral channels in fused silica glass with a channel width of 74 μm. The channels were filled with dyes (see inset, scale: 140 μm). **e** Polymeric microstructures of an out-of-plane mixer structure (scale: 600 μm). **f** Microfluidic mixer structure in fused silica glass with a channel width of 74 μm (scale bar: 280 μm). As can be seen the 3D structures can be replicated with high fidelity and no deformations

sintered fused silica glass shows a hydrophilic surface (see Supplementary Figure 3) and the same surface energy of ~60 mN m$^{-1}$ like commercial fused silica glass[21–23]. The surface properties of fused silica glass can be adjusted for biofunctionalization using for e.g., silanization[33].

A significant benefit of this nanocomposite approach is that it is compatible with different materials and fabrication processes for the sacrificial templating component. Simple microfluidic channels were fabricated by immersing nylon threads in the nanocomposite (see Fig. 1b). Multiple threads were connected by thermally fusing the threads under light pressure at 100 °C. More complex two-dimensional microfluidic channels were fabricated

by microlithography using PEGDA as material for the template (see Fig. 1c).

**Templates produced by melt electrowriting.** Another accessible additive manufacturing approach for the templates is melt electrowriting, which creates continuous, ultrafine diameter fibers[34,35]. The smooth, uniform fibers produced in this process are well-suited as templates for generating structures such as complex microfluidic channel networks. Figure 1d shows a three-dimensional mesh structure of well-stacked melt electrowritten fibers. The mesh was completely immersed in the nanocomposite and processed/sintered as a bulk structure to give suspended

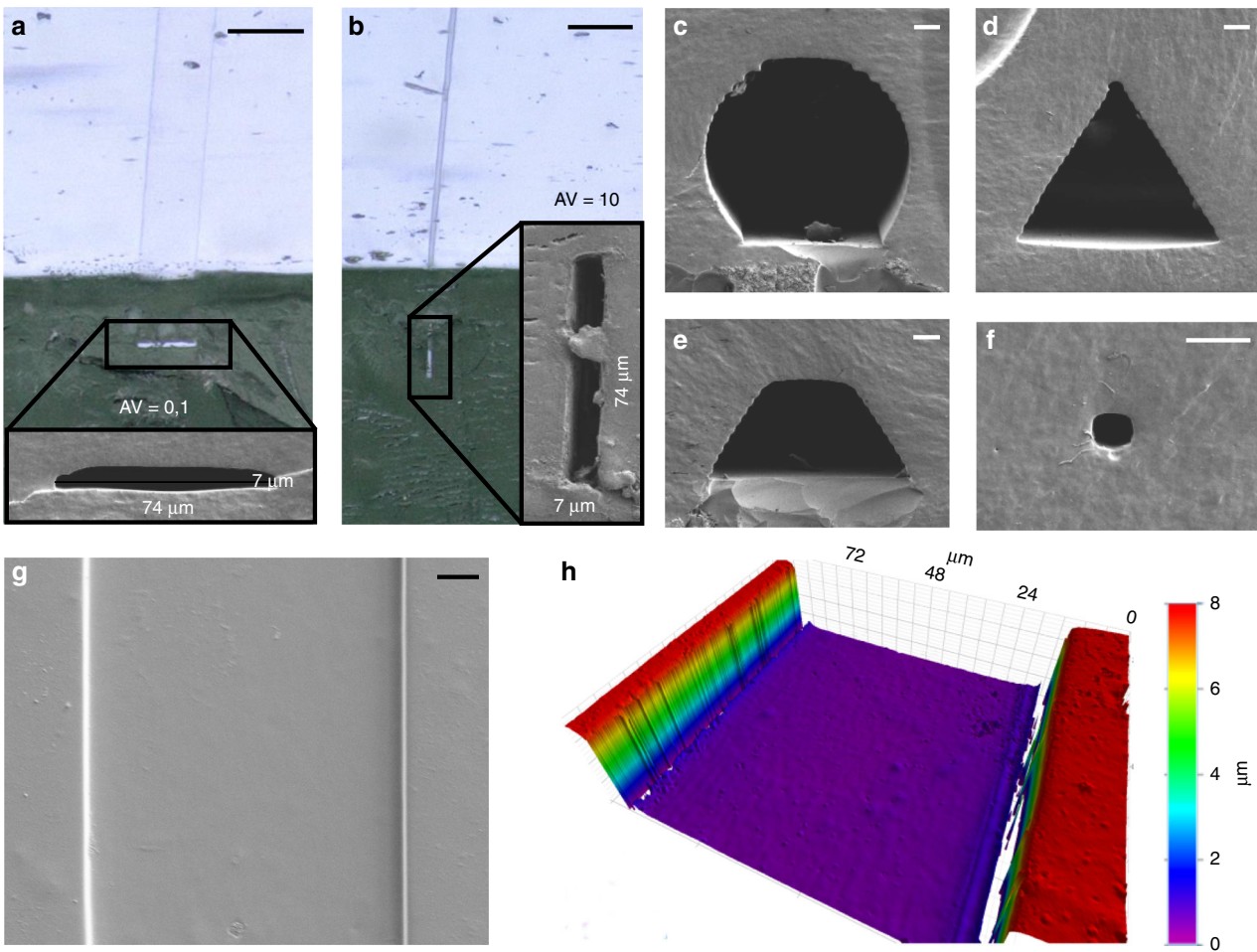

**Fig. 3** Characterization of suspended hollow microstructures in fused silica. **a**, **b** SEM of rectangular channel cross-section with an aspect ratio of 0.1 and 10 (scale: 100 μm). **c–f** SEM of spherical, triangular, trapezoidal, and rectangular channel cross-sections (scale: 10 μm). All templates were fabricated using direct laser writing. The flattened side of the "spherical" channel cross-section is due to the 2-photon polymerization 3D printing process of the template. Structures are printed on glass slides and a certain contact area is required to prevent the structure from detaching from the glass. **g**, **h** SEM and white light interferometry of the channel structure from **a** with a mean roughness of Ra ~20 nm (scale: 10 μm)

channels in one step. The inverse micromesh structure obtained in fused silica glass is shown in Fig. 1e. The glass cavities have a diameter of 18.4 μm.

**Templates produced by direct laser writing**. In order to generate highly complex template microstructures direct laser writing was used. Figure 2 shows exemplary microstructures fabricated using this approach. The feature resolution achievable with this process is unmatched and comparably complex microstructures have never been manufactured in fused silica glass.

As an example, we generated a DNA double-helix structure (Fig. 2a, b), intertwined microfluidic spiral channels (Fig. 2c, d) and a three-dimensional microfluidic channel structure with out-of-plane geometries (Fig. 2e, f).

Using this approach, microfluidic channel structures with low and high aspect ratios can be fabricated with ease. Figure 3a, b shows two rectangular cross sections with an aspect ratio of 0.1 and 10, respectively. Furthermore, the cross sections of the channel structures can be changed to arbitrary shapes: Fig. 3c–e shows circular, trapezoid, and triangular channel cross-sections with a maximum feature size of 74 μm. Channels of exceptionally small sizes can be fabricated as shown exemplary in Fig. 3f, showing a microchannel of $7 \times 7$ μm$^2$ cross section. While the channel cross-sections are in the range of only a few micrometers, the lengths of the channels are in the range of centimeters.

Different strategies have been described to connect microfluidic chips to macroscopic pumps (world-to-chip interface). We have connected the chips to a pump using 3D-printed connectors (see Supplementary Figure 2a, b). Using this interconnection, we demonstrate a hydrodynamic flow focussing experiment in a four-inlet, one-outlet flow focussing channel. Hydrodynamic flow focussing is a key factor in microfluidic synthesis of functional materials and supramolecular self-assemblies[18,36].

The surface roughness of the produced fused silica depends on the type of mold used, i.e., the roughness of the polymeric template structure. The exceptionally smooth surfaces of the channels resulting from sacrificial templates fabricated by laser direct writing in fused silica glass is shown in Fig. 3g, h showing mean roughness of Ra ~20 nm. Surfaces of optical quality are therefore achievable. The roughness of the glass part depends on the roughness of the used template structure. We have shown in our previous work that using smooth glass surfaces with a surface roughness of Rq ~2–3 nm can be fabricated[22,23].

In summary we described a novel potent technique to fabricate, with high accuracy, arbitrary embedded freeform three-dimensional suspended hollow microstructures in transparent fused silica glass by using a sacrificial template replication

process. This technique will enable numerous applications in flow-through synthesis and analysis, microfluidics, and Lab-on-a-Chip devices for chemical miniaturization as well as applications in optics and photonics.

## Methods

**Materials**. Amorphous silica nanopowder of type Aerosil OX50 was kindly provided by Evonik, Germany. Hydroxyethylmethacrylate (HEMA) was purchased from Alfa Aesar, Germany. Tetraethylenglycoldiacrylate (TEGDA), poly-ethylenglycoldiacrylate 550 (PEGDA-550), phenylbis(2,4,6-trimethylbenzoyl) phosphine oxide, propylene glycol methyl ether acetate (PGMEA), and 2,2-dimethoxy-2-phenylacetophenone (DMPAP) were purchased from Sigma–Aldrich. Negative-tone photoresists IP-S was purchased from Nanoscribe, Germany. 2-propanol was purchased from Carl Roth, Germany.

**Direct laser writing**. Prior to usage in the fabrication process, glass substrate (25 × 25 × 0.7 mm, from Nanoscribe GmbH, Germany) were activated by oxygen plasma in order to enhance the adhesion of the photoresist to the glass. Note that additional silanization is not recommended since the bonding was found to be too strong to detach the polymer from the substrate when embedded in the nanocomposite. The 3D objects were fabricated using a commercial lithography system Photonic Professional GT (Nanoscribe GmbH, Germany). Negative-tone photoresist IP-S was used as photoresist and was drop-casted on the activated substrate. The writing speed was set to 100 mm s$^{-1}$ with the slicing distance set to 1 μm ("IP-S recipe" in software Describe, Nanoscribe GmbH, Germany). The numerical aperture of the objective lens is NA = 0.8 with an effective working distance of 400 μm. Both, solid writing and core-shell approach have been explored successfully. After exposure, the sample was developed in PGMEA for 10 min and rinsed with another bath of PGMEA for 30 s.

**Lithography**. PEGDA-550 was blended with 0.5 m% of the photoinitiator phenylbis(2,4,6-trimethylbenzoyl)phosphine oxide. PEGDA-550 was then structured using a custom-built lithography system based on a digital mirror device (DMD)[49]. Structuring was done at a wavelength of 365 nm for 28 s at an exposure intensity of 2.6 mW cm$^{-2}$. After the exposure the polymeric structures were developed in 2-propanol for 30 s.

**Transfer of microstructures**. The microstructures were fabricated on a glass slide and the nanocomposite was cast on top. After polymerization, the nanocomposite with the embedded microstructure was peeled off the glass and the open structure was sealed with a second layer of nanocomposite. The process is shown in Supplementary Figure 1. The bonding strength of the nanocomposite was higher than 400 kPa (tested using tensile testing) allowing to comfortably handling the polymerized nanocomposites. After sintering a bulk glass component without any internal interface is obtained.

**Melt electrowriting**. PCL (PC-12, Corbion, the Netherlands) was used as received and processed using a custom-built melt electrowriting printer[50]. One gram of PCL was placed in an electrically heated (75 °C) syringe and pneumatically delivered to a 23 G nozzle using air (1.0 bar). This nozzle is positioned 6 mm above a collector, and a total of 5.5 kV is applied across this collector distance. Direct writing was performed using x/y linear stages and samples were used as sacrificial templates without post-processing.

**Preparation of the nanocomposite**. The nanocomposite used in this work consisted of 68 vol% HEMA, 7 vol% of TEGDA, and 25 vol% of POE, which were mixed prior to the dispersion process[23]. Afterwards 40 vol% Aerosil OX50 were dispersed in the monomeric mixture. The nanopowders were added in small increments to this mixture using a laboratory dissolver (R 130, IKA, Germany). Afterwards 0.5 m% (referred to the amount of reactive monomer) of the photoinitiator DMPAP was added following a further dispersion step of 30 min. Entrapped air bubbles were removed using a desiccator and a vacuum pump.

*Embedding and polymerization of the nanocomposite*: For embedding of the polymer filaments into the nanocomposites the latter were heated to 60 °C prior to the casting process. This reduces the risk of entrapping air bubbles. The nanocomposites were subsequently polymerized at a wavelength of 300–400 nm at an exposure intensity of 12 mW cm$^{-2}$ for 2 min.

**Heat treatment**. Thermal debinding was done using an ashing furnace (type AAF, Carbolite/Gero, Germany). Sintering was done using a tube furnace (type STF16/450, Carbolite/Gero, Germany) at a temperature of 1300 °C and a pressure of 5 × 10$^{-2}$ mbar with a heating rate of 3 K min$^{-1}$. The parameters for thermal debinding and sintering can be found in Supplementary Table 1.

*Fabrication of 3D-printed clamp*: The designed clamp for connecting tubing to the glass chip was manufactured using a 3D printer (ProJet MJP 2500/2500 Plus, 3D systems, U.S.) from VisiJet® M2R-CL resin.

**Assembly of 3D-printed clamp with glass chip**. In order to connect inlets and outlet with microfluidic tubing (Tefzel™ (ETFE) Tubing 1/16" outer diameter and 0.040" inner diameter, IDEX Health & Science, LLC, USA), the glass chips were assembled with the top and bottom part of 3D-printed clamp using M2 bolts. O-rings (inner diameter 0.7 mm, thread diameter 1 mm) were used to seal the inlet and outlet holes of the glass chip to the 3D-printed clamp. Microfluidic tubings were directly inserted inside the holes (diameter 1.5 mm) on the top clamp to connect inlet and outlets of the glass chip.

**Hydrodynamic flow focusing experiments**. A concentration of 10 μM fluorescein sodium salt (Sigma–Aldrich Chemie GmbH, Switzerland) was prepared in deionized water and pumped through the glass microfluidic channel's two middle inlets using a syringe pump (neMESYS 290 N, CETONI GmbH, Germany), whereas the outer two inlets were probed with deionized water as sheath flows. The flow rate ratio (FRR, i.e., the total flow rate of sheath flows divided by total flow rate of middle streams) was tuned during the experiments. Supplementary Figure 2c and d show, respectively, fluorescence microscopy images corresponding to FRR = 0 and FRR = 1.

**Fluorescence microscopy setup**. The images were taken using a microscope (Nikon Eclipse Ti, Nikon GMBH, Switzerland) equipped with a 20X objective (CFI S Plan Fluor ELWD 20XC, Nikon GmbH, Switzerland) and a colour camera (RETIGA R1, Rochester, USA). A LED light source (Omicron LedHUB, Omicron-Laserage Laserprodukte GmbH, Germany) with a wavelength of 470 nm was used for illumination with a power of 30 mW.

## Data availability

The authors declare that the data supporting the findings of this study are available within the paper and its Supplementary Information files. All other relevant data are available from the corresponding author upon reasonable request.

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

## Acknowledgements

This work has been partly funded by the German Federal Ministry of Education and Research (BMBF), "Fluoropor", funding code: 03 × 5527. We thank Evonik for providing nanopowders. We thank Richard Thelen for white light interferometry. This work was partly carried out with the support of the Karlsruhe Nano Micro Facility (KNMF) (www.kit.edu/knmf), a Helmholtz Research Infrastructure at Karlsruhe Institute of Technology (KIT) (www.kit.edu). S.S. and J.P.L. acknowledge funding from the European Research Council (ERC-2015-STG No. 677020) and the Swiss National Science Foundation (project no. 200021_160174).

## Author contributions

F.K. and B.E.R. conceived the idea. F.K designed the experiments, synthesized the material, and wrote the manuscript. P.R. and K.A. performed the replication process. A.Q. and M.T. performed direct laser writing. J.P. and S.S. CAD Design of microfluidic chips and fluidic experiments. A.H. and P.D.D. performed melt electrowriting. D.H. performed lithography experiments. B.E.R. and D.H. contributed to the writing of the paper.

## Additional information

**Competing interests:** The authors declare no competing interests.

