## [Peer Review File · Nature Communications]

Reviewers' comments:

Reviewer #1 (Remarks to the Author):

The authors report on fabrication of suspended 3D microfluidic structures in fused silica using sacrificial template replication. The idea is not completely new whilst the technique developed by the authors is attractive in the sense that large scale 3D microfluidic systems of complex geometries have been demonstrated. In my opinion, the manuscript may be considered for publication in Nat. Commun. Nevertheless, a few comments/questions should be clarified before such a decision would be made.

1. Line #36, "As of today there is no method for generating truly arbitrary three-dimensional hollow structures of centimeter lengths and few micrometers diameter in bulk fused silica glass".

This statement is wrong. Actually, it has been demonstrated before that centimeter-long microchannels of micrometer-scale diameters can be realized in a porous glass using femtosecond laser micromachining [C. Liu et al, JLMN-Journal of Laser Micro/Nanoengineering 8, 170-174 (2013)].

The statement should be modified and the previous result should be discussed to compare with the result in the current paper.

2. How about the mechanical strength of the fused silica obtained after the sintering of the nanocomposite? Have the authors compared it with bulk fused silica?

3. How about the bonding strength when sealing the open structure with the nanocomposite?

4. Have the authors compared the costs of the fabrication approaches using nanocomposite and bulk fused glass?

5. In the femtosecond laser direct writing, what are the parameters (such as NA, working distance, etc) of the focal system?

6. Fig. 3c, why the so-called "spherical" cross section does not have a completely circular shape?

7. In the transfer of the microstructure, would the 3D polymer microstructure deform? How significant is the deformation?

Reviewer #2 (Remarks to the Author):

This manuscript describes the fabrication of arbitrary three-dimensional suspended hollow microstructures in transparent fused silica glass using a sacrificial template replication strategy. The manuscript also covers a few state-of-the-art techniques for fabricating microstructures in fused silica glass and discusses their limitations, thereby providing a useful perspective for the work. The work is complementary to a paper recently published in Nature on 3D printing of transparent fused silica glass because it allows to create hollow microstructures in fused silica with high resolution and arbitrary shapes. The results are convincing and the work can be of interest to Nature Communications' readership. However, the manuscript would need some improvements before publication.

Major comments:

Applications. Applications were the key motivation for this work but applications per se are not shown in the manuscript. Filling/using the produced structures is missing and the manuscript is essentially a collection of images of structures. What about the world-to-chip challenge? For example, how would the structures shown in Fig. 2D be connected to tubings or input/output channels?

Fugitive inks. The sacrificial template replication approach described in the manuscript resembles the fugitive ink method published earlier (Lewis et al.). Can the authors elaborate more on the difference between both approaches?

More quantitative analysis of resolution and roughness. The authors mention that high resolution structures can be obtained in polymers. Indeed, hot embossing and injection molding can yield structures with micrometers precision and very smooth surfaces (on the nanometer scale). Do the authors have more results than what is shown in Figure 3, which is limited to a few examples and where it seems that some imperfections are present. An "exceptional smooth surface" is also mentioned in the text (RMS roughness of ~20 nm). Such a roughness is in fact relatively high, in my opinion. I also do not understand what are the criteria used to term such surfaces "of optical

quality". Can the authors provide a more systematic analysis of the resolution and roughness, and in particular describe what were the target dimensions of the formed structures and what was the experimentally obtained dimensions. Are there any special design rules (rounding of corners, minimum diameter, minimum achievable radius for curved structures, etc).

Surface chemical functionalization. Creating structures and scaffolds sometimes need to be followed with surface functionalization or bio-functionalization. Was there any surface treatment implemented by the authors? Would depletion of reactants be an issue, for example for the treatment of structures as shown in Fig. 1E?

Demonstrating applications. Finally, the work would considerably benefit from showing one or a few applications.

Minor comments:

(1) Table S2 is mentioned on line 90. Is it rather Table S1?

(2) Caption of Figure 1A: how was the polymeric template removed?

Reviewer #3 (Remarks to the Author):

In this paper, the author proposed an approach to produce complex, suspended hollow microstructures in fused silica glass. The process combines the concept of sacrificial template replication with a room-temperature molding process for fused silica glass. The template molding can be achieved by using nylon threads, poly (eth 79 ylene glycol diacrylate) (PEGDA) scaffolds, poly (ϵ -caprolactone) (PCL) microfiber meshes produced by melt electrowriting and complex polymeric microstructures fabricated by direct laser writing. The molding process for fused silica glass based on the nanocomposite approach is compatible with different materials and fabrication processes for the sacrificial templating component. The author declares that it is possible to generate nearly arbitrarily-shaped freeform three dimensional channels and hollow structures within fused silica glass. The actual results are very interesting and the paper is well presented. It is suggested to be accepted for publication after addressing the follow questions.

1. Channels of exceptionally small sizes can be fabricated as shown exemplary in Figure 3F, showing a microchannel of $7 \times 7 \mu\text{m}^2$ cross section. What is the microchannel size tolerance after repeating fabrication of the microchannel with the same size?

2. The author declares that the surface roughness of the produced fused silica depends on the type of mold used, i.e., the roughness of the polymeric template structure. Does the sintering process at 1300 °C have more influence on the surface roughness of the hollow microstructures in fused silica glass? Does it have the influence on the geometry of the hollow microstructures?

Dear reviewers,

Please find enclosed our response to the comments of the reviewers. We thank the reviewers for their helpful suggestions. In the following we describe the changes made to the manuscript. The reviewers' comments are set in upright font whereas our comments are set in italic. All changes to the manuscript and supplementary information are marked in yellow.

Reviewer #1

1. Comment 1. Line #36, "As of today there is no method for generating truly arbitrary three-dimensional hollow structures of centimeter lengths and few micrometers diameter in bulk fused silica glass". This statement is wrong. Actually, it has been demonstrated before that centimeter-long microchannels of micrometer-scale diameters can be realized in a porous glass using femtosecond laser micromachining [C. Liu et al, JLMN-Journal of Laser Micro/Nanoengineering 8, 170-174 (2013)]. The statement should be modified and the previous result should be discussed to compare with the result in the current paper.

Answer1: We disagree with the reviewer's statement. We have claimed that the formation of such channels in fused silica has not yet been achieved. The work cited by the reviewer is not showing the formation of channels in fused silica, but in a borosilicate glass. In this technique, femtosecond laser writing is only effective, because the glass is porous due to removal of a borate rich phase from phase-separated alkali-borosilicate glasses. After structuring with the laser, the pores are removed at 1130-1150 °C. This strategy is not applicable to fused silica glass and the resulting glass is, as the authors correctly state, "high silica glass" not fused silica. The glass composition is given as 95.5% SiO₂-4% B₂O₃-0.5Na₂O (wt. %). The authors further cite the following paper as source of fabricating the porous glass: <https://doi.org/10.1063/1.1946897> where it is stated that "the analytical composition of the porous glass obtained is 97.0% SiO₂-2.1% B₂O₃-0.8% Al₂O₃-0.05% Na₂O-0.05% CaO".

Besides the fact that the method of Liu et al. cannot be applied to fused silica glass, it has additional drawbacks: The channel shape created by the laser cannot be effectively controlled. In an earlier work of the same group (Liao et al., OPTICS LETTERS, 35 (19), 2010), the authors claim that "the channels are much larger and more symmetric than the focal spot, probably because of the homogenous thermal conductivity of porous glass, which results in the nearly isotropic ablation. Nevertheless, a small asymmetry of cross-section shape of the microchannel could still reflect the elliptical focused spot". Liu et al. (JLMN, 8(2), 2013), claim that "the diameter of the microchannel decreases with decreasing pulse energy. Thus, tuning pulse energy of the femtosecond laser can be an efficient approach for controlling the cross sectional size of the microfluidic channel". However, it remains unclear how the channel size is controlled and if the channels can be produced with a specific cross-section. Moreover, the channel cross-sections vary greatly with different channel diameters (see Liu et al, JLMN, 8(2), 2013, figure 2 b) and they are neither all elliptical, nor circular. Probably due to non-isotropic shrinkage of the parts, some channels display a very thin hairpin on top of the channel, which remains uncommented by the authors. Also, the method reported by Liu et al., just like other femto- or nanosecond laser ablation methods, suffers from residual debris in the channel structures. In the paper the authors have scanned the channel several times with the laser to remove residual debris, which is only partly successful, see for instance Liu et al, OPTICS Express, 20 (4), 4291, figure 4, that displays the coarse walls of the channels (SEM) or Liu et al. JLMN 8 (2), 2013, figure 2 a, which displays light scattering on the coarse structure of the channels, as does figure 3d and 4d in Cheng et al, "how to make a 3D micromixer" DOI: 10.1117/2.1201202.004150.

Taken together, we stand by our claim of our method being the first for generating truly arbitrary three-dimensional hollow structures of centimetre length and few micrometers diameter in bulk

fused silica glass. We have added the following section to the manuscript to acknowledge the work of the Midorikawa group:

“A method for three-dimensional structuring of high-silica glasses with an SiO₂ content of 95 % - 97 % via femtosecond laser writing has been previously reported, using a porous glass similar to VYCOR.¹⁴ The structures produced however display a coarse wall structure and non-uniform channel cross-sections.”

2. How about the mechanical strength of the fused silica obtained after the sintering of the nanocomposite? Have the authors compared it with bulk fused silica?

Answer 2: We have intensively studied the mechanical strength of the sintered fused silica glass in comparison to commercially available fused silica glass. In all tests, the results obtained from our glasses were identical to the values obtained for commercial fused silica glass. We have published these results in previous papers. We have added a section explaining that the sintered fused silica glass parts are chemically and physically indistinguishable from commercial fused silica glass for making this clear for the reader:

“We have recently demonstrated that the sintered fused silica glass parts show the same high optical transparency in the UV, visible and infrared region as well as the same mechanical, chemical and thermal stability. We further demonstrated that the sintered fused silica glass show a hydrophilic surface (see Figure S3) and the same surface energy of ~60 mN/m like commercial fused silica glass.^{21-23.”}

3. How about the bonding strength when sealing the open structure with the nanocomposite?

Answer 3: The two layers were polymerized using radical polymerization resulting in a covalent bond between both layers. This is not a classical bonding procedure. This process results in one bulk nanocomposite component. We have performed tensile testing of this interface by applying tensile stress of up to 400 kPa between two polymerized nanocomposites blocks (diameter = 2 cm, height = 1 cm). We were unable to break this interface without breaking the bulk material. After the sintering process the process results in one single bulk glass component which shows no interface. The mechanical properties of the sintered parts are identical to commercial fused silica glass (see Answer 2). We have added the following sentence to further clarify this for the reader:

“The bonding strength of the nanocomposite was higher than 400 kPa (tested using tensile testing) allowing comfortably handling the polymerized nanocomposites. After sintering a bulk glass component without any internal interface is obtained.”

4. Have the “authors compared the costs of the fabrication approaches using nanocomposite and bulk fused glass?

Answer 4: As we have pointed out Answer 1, there is no other method for producing three-dimensional, smooth, centimetre long channels in fused silica glass. Relevant reference methods for fused silica glass structuring produce coarse channel walls or tapered channels. The powders used for making the nanocomposites are side-products of the production of optical fibres and are commercially available at around 5 Euro per kg. Including organic binder, the price is about 7.30 Euro in total for the nanocomposite required for generating 1 kg of fused silica glass. However, as these

values are strongly dependent on the purchased batch size and batch volume, we do not feel comfortable stating these values as representative.

5. In the femtosecond laser direct writing, what are the parameters (such as NA, working distance, etc) of the focal system?

Answer 5: We thank the reviewer for this comment as this data is indeed missing. We have added the missing information to the manuscript.

"The numerical aperture of the objective lens is $NA = 0.8$ with an effective working distance of $400 \mu\text{m}$."

6. Fig. 3c, why the so-called "spherical" cross section does not have a completely circular shape?

Answer 6: The "spherical" cross section does not have a completely circular shape as the polymer structure was printed on a flat glass substrate using 2-photon polymerization. We have added the following section to the manuscript:

"The flattened side of the "spherical" channel cross-section is due to the 2-photon polymerization 3D printing process of the template. Structures are printed on glass slides and a certain contact area is required to prevent the structure from detaching from the glass."

7. In the transfer of the microstructure, would the 3D polymer microstructure deform? How significant is the deformation?

Answer 7: We used highly crosslinked acrylates as template structures which show very a high dimensional stability. The 3D polymer microstructures showed no deformation during the transfer process. As can be seen in Figure 2, several intricate structures have been embedded into fused silica glass and show no distortion. We have added the following sentence to Figure 2:

"As can be seen the 3D structures can be replicated with high fidelity and no deformations."

Reviewer #2

8. Applications. Applications were the key motivation for this work but applications per se are not shown in the manuscript. Filling/using the produced structures is missing and the manuscript is essentially a collection of images of structures. What about the world-to-chip challenge? For example, how would the structures shown in Fig. 2D be connected to tubings or input/output channels?

Answer 8: We have shown the filling of exemplary channels with dyed water (see Figure 2D, zoom). We have added a hydrodynamic flow-focussing experiment to the supplementary material, which also tackles the world-to-chip-challenge. Figure S2 shows a flow focussing chip with a 3D printed chip-to-world clamp connector which we have also used during the experiments displayed in the main test. We have added a sentence in the manuscript referring to the section in the supporting materials:

“Different strategies have been described to connect microfluidic chips to macroscopic pumps (world-to-chip interface). We have connected the chips to a pump using 3D printed connectors (see supporting information Figure S2a/b). Using this interconnection, we demonstrate a hydrodynamic flow focussing experiment in a four-inlet, one-outlet flow focussing channel. Hydrodynamic flow focussing is a key factor in microfluidic synthesis of functional materials and supramolecular self-assemblies.”^{18,36}

We have added Figure S2 to the supplementary information.

Figure S2. (a) Schematic illustration of 3D printed clamp designed for connecting microfluidic tubing to a four-inlet and one-outlet microfluidic glass chip. (b) Picture of the four-inlet and one-outlet microfluidic glass chip after assembling the 3D printed clamp and connecting the microfluidic tubings. (c/d) Fluorescence microscopy images showing hydrodynamic flow focusing of fluorescein dye using four-inlet and one-outlet microfluidic glass chip where the two sheath flows (outer inlets) are water and the middle streams (middle inlets) are solutions of fluorescein dye in deionized water (10 μ M). Flow rate ratio (FRR), calculated as total flow rate of sheath flows divided by total flow rate of middle streams is $FRR=0$ and $FRR=1$, respectively in (c) and (d).

We have added the following sections to the Materials and Methods section:

Fabrication of 3D printed clamp: The designed clamp for connecting tubing to the glass chip was manufactured using a 3D printer (ProJet MJP 2500/2500 Plus, 3D systems, U.S.) from VisiJet® M2R-CL resin.

Assembly of 3D-printed clamp with glass chip: In order to connect inlets and outlet with microfluidic tubing (Tefzel™ (ETFE) Tubing 1/16" outer diameter and 0.040" inner diameter, IDEX Health & Science, LLC, USA), the glass chips were assembled with the top and bottom part of 3D-printed clamp using M2 bolts. O-rings (inner diameter 0.7 mm, thread diameter 1 mm) were used to seal the inlet and outlet holes of the glass chip to the 3D-printed clamp. Microfluidic tubings were directly inserted inside the holes (diameter 1.5 mm) on the top clamp to connect inlet and outlets of the glass chip.

Hydrodynamic flow focusing experiments: A concentration of 10 μM fluorescein sodium salt (Sigma-Aldrich Chemie GmbH, Switzerland) was prepared in deionized water and pumped through the glass microfluidic channel's two middle inlets using a syringe pump (neMESYS 290N, CETONI GmbH, Germany), whereas the outer two inlets were probed with deionized water as sheath flows. The flow rate ratio (FRR, i.e., the total flow rate of sheath flows divided by total flow rate of middle streams), was tuned during the experiments. Figure S2(c) and (d) show respectively, fluorescence microscopy images corresponding to $\text{FRR}=0$ and $\text{FRR}=1$.

Fluorescence microscopy setup: The images were taken using a microscope (Nikon Eclipse Ti, Nikon GMBH, Switzerland) equipped with a 20X objective (CFI S Plan Fluor ELWD 20XC, Nikon GmbH, Switzerland) and a colour camera (RETIGA R1, Rochester, USA). A LED light source (Omicron LedHUB, Omicron-Lasertechnik Laserprodukte GmbH, Germany) with a wavelength of 470 nm was used for illumination with a power of 30 mW.

9. Fugitive inks. The sacrificial template replication approach described in the manuscript resembles the fugitive ink method published earlier (Lewis et al.). Can the authors elaborate more on the difference between both approaches?

Answer 9: Lewis et al. described the so-called fugitive ink method where a template material is embedded in a polymer and removed by liquefying the template structure at an elevated temperature. The major advantage of our technique is that the templates are removed in the gas phase during the thermal debinding process. There is no risk of material redeposition or channel blocking by incomplete removal. In addition, diffusion limitations also do not apply to the removal process in the gas phase.

We added an additional sentence to the manuscript explaining the difference of the template removal compared the approach by Lewis et al. (The paper was already included as reference in the original manuscript):

"This is an advantage over many sacrificial template replication process like the fugitive ink technology where the template has to be liquefied and washed out of the bulk material.³² As the templates are removed in the gas phase, there is no material redeposition or channel blocking by incomplete removal. In addition, diffusion limitations for the removal of the template do not apply."

10. The authors mention that high resolution structures can be obtained in polymers. Indeed, hot embossing and injection molding can yield structures with micrometres precision and very smooth surfaces (on the nanometer scale). Do the authors have more results than what is shown in Figure 3, which is limited to a few examples and where it seems that some imperfections are present. An "exceptional smooth surface" is also mentioned in the text (RMS roughness of ~ 20 nm). Such a roughness is in fact relatively high, in my opinion. I also do not understand what are the criteria used to term such surfaces "of optical quality".

Answer 10: The industry standard for injection moulding defines the highest possible surface finish with super high glossy, which is between 12 and 25 nm Ra (see Society of Plastic Industry SPI standards, see for e.g. *Innovation Trends in plastics Decoration and Surface Treatment*, Edward B. Crutchley, Smithers Rapra Technology, 2014). Our technique is also within the range of roughness provided by glass moulding techniques, i.e. precision glass moulding, where 10-20 nm are typically accepted surface roughness values for lenses (see, e.g., Zhang & Liu *Front. Mech. Eng.* 12 (1), 3-17, 2017). These values lead us to conclude that the value we achieve is in fact not very high and adequate for applications in optics. Typical roughness values for two-photon polymerization as performed by the Nanoscribe system are (according to the company) in the range of 10 nm Ra. Takada et al., *Applied physics letters* 86, 071122, 2005 also claim surface roughness values of 4-11 nm for direct laser writing. Please note that these are Ra values, the Rq values are usually slightly higher. Given this initial roughness of the molds the moulded surfaces will obviously possess higher Rq values as moulding cannot reduce surface roughness. We therefore reason that the achievable surface roughness are largely due to the roughness of the mould. We have previously shown that the roughness of the sintered glass parts depends on the roughness of the mold against which the nanocomposite is polymerized. We have shown in intensive studies that using smooth glass moulds exceptionally smooth glass surfaces with a roughness of Rq ~ 2-3 nm can be fabricated. (Kotz et al., 544, 337-339, *Nature*, 2017; Kotz et al., 30, 1707100, *Advanced Materials*, 2018).

In direct comparison with relevant reference methods for glass structuring (see also Answer 1) using femtosecond laser writing, which produce surface roughness of ~120 nm (see He et al, *Appl Phys B*, 105:379-384,2011) the surfaces we have shown are in fact exceptionally smooth.

"The roughness of the glass part depends on the roughness of the used template structure. We have shown in our previous work that using smooth glass surfaces with a surface roughness of Rq ~ 2-3 nm can be fabricated."^{22,23}

11. Can the authors [...] describe what were the target dimensions of the formed structures and what was the experimentally obtained dimensions.

During the sintering process the parts shrink isotopically in independence of the solid loading. The equation for the calculation of the linear shrinkage is:

$$Y_s = 1 - (\Phi/(\rho_f/\rho_t))^{1/3}$$

With solid loading Φ , theoretical density ρ_t and final density ρ_f . Using this equation the theoretical shrinkage for a nanocomposite with a solid loading of 40 vol% corresponds to a linear shrinkage of 26.3 %. We have added the following sections to the main part of the manuscript:

"During the sintering process the parts shrink isotopically in dependence of the solid loading. Here we used a nanocomposite with a solid loading of 40 vol% resulting in a linear shrinkage of 26.3 %. For example the length of the upper channel of the mixer in Figure 2E showed the expected linear shrinkage of 26.28 % from 2121.95 μm to 1564.23 μm . Further information on the shrinkage calculation can be found in the supplementary information."

We have added the following text to the supporting information:

"The shrinkage during the heat treatment is isotropic and can be calculated in dependence of the solid loading using the following formula:

$$Y_s = 1 - (\Phi/(\rho_f/\rho_t))^{1/3}$$

Φ – solid loading

ρ_f – final density

ρ_t – theoretical density”

12. Are there any special design rules (**rounding of corners, minimum diameter, minimum; achievable radius for curved structures, etc**).

As stated, the shrinkage is isotropic so the structures will shrink in size. There are no special design rules.

13. Surface chemical functionalization. Creating structures and scaffolds sometimes need to be followed with surface functionalization or bio-functionalization. Was there any surface treatment implemented by the authors? Would depletion of reactants be an issue, for example for the treatment of structures as shown in Fig. 1E?

As demonstrated, the glass generated by our process is chemically and physically indistinguishable from commercial fused silica glass, and hydrophilic. We have added a figure to the supplementary information showing the hydrophilicity of our glass with a static water contact angle of 30 °. In a previous publication (In Kotz et al., Advanced Materials 28, 4646-4650, 2016) we demonstrated that the surface energy of the sintered fused silica glass is the same as for commercial fused silica glass. Therefore any protocol developed for the (bio)chemical functionalization of fused silica glass can also be applied to our material.

We have added the following the section to the manuscript:

“We have recently demonstrated that the sintered fused silica glass parts show the same high optical transparency in the UV, visible and infrared region as well as the same mechanical, chemical and thermal stability. We further demonstrated that the sintered fused silica glass show a hydrophilic surface (see Figure S3) and the same surface energy of ~60 mN/m like commercial fused silica glass.²¹⁻²³ The surface properties of fused silica glass can be adjusted for biofunctionalization using for e.g. silanization.³³”

We have added the following figure to the supplementary information:

Figure S3. Contact angle measurement: Sintered glass parts possess hydrophilic surface properties. 5 μ l water droplet on a sintered glass surface showing a low contact angle of around 32 °.

14. Demonstrating applications. Finally, the work would considerably benefit from showing one or a few applications.

We have included an application for this technology, see Answer 8.

We have added the following sentence to the manuscript:

“Different strategies have been described to connect microfluidic chips to macroscopic pumps (world-to-chip interface). We have connected the chips to a pump using 3D printed connectors (see supporting information Figure S2a/b). Using this interconnection, we demonstrate a hydrodynamic flow focussing experiment in a four-inlet, one-outlet flow focussing channel. Hydrodynamic flow focussing is a key factor in microfluidic synthesis of functional materials and supramolecular self-assemblies.^{18,36}”

15. Table S2 is mentioned on line 90. Is it rather Table S1?

We thank the reviewer for this comment. We corrected the number in the manuscript.

16. Caption of Figure 1A: how was the polymeric template removed?

The polymeric template is removed during the thermal debinding process. We have changed the sentence in the caption of Figure 1A:

“The polymeric template is removed during the thermal debinding process and leaves the according hollow cavity.”

Reviewer #3

17. Channels of exceptionally small sizes can be fabricated as shown exemplary in Figure 3F, showing a microchannel of $7 \times 7 \mu\text{m}^2$ cross section. What is the microchannel size tolerance after repeating fabrication of the microchannel with the same size?

Answer 17: We have fabricated three equal microchannel templates and replicated them into glass microchannels to demonstrate that this process allows fabricating microchannels with high reproducibility. Each channel diameter was measured three times. The resulting microchannels had a diameter of $123.87 \pm 0.16 \mu\text{m}$, $124.23 \pm 0.51 \mu\text{m}$ and $124.13 \pm 0.49 \mu\text{m}$ demonstrating the high reproducibility for the replication process. We have added the following sentence to the manuscript.

"To demonstrate that this process can be used to fabricate microchannels with high reproducibility we have fabricated three microchannels using the same protocol. The final microchannels had a diameter of $123.87 \pm 0.16 \mu\text{m}$, $124.23 \pm 0.51 \mu\text{m}$ and $124.13 \pm 0.49 \mu\text{m}$ each measured three times."

18. The author declares that the surface roughness of the produced fused silica depends on the type of mold used, i.e., the roughness of the polymeric template structure. Does the sintering process at $1300 \text{ }^\circ\text{C}$ have more influence on the surface roughness of the hollow microstructures in fused silica glass?

*Answer 18: We have shown in our previous studies that the parts must be sintered in a very narrow process window around $1300 \text{ }^\circ\text{C}$ in order to achieve highly transparent fused silica glass (Kotz et al. *Advanced Materials*, 28, 4646-4650, 2016). Below $1300 \text{ }^\circ\text{C}$ the parts remain porous and non-transparent. Above $1300 \text{ }^\circ\text{C}$ the material crystallizes to the non-transparent cristoballite. The sintering itself does not influence the surface roughness significantly once the part is sintered to full density. We have discussed the roughness of the parts in detail in the answer to question 10.*

19. Does it have the influence on the geometry of the hollow microstructures?

Answer 19: This question has also been raised by reviewer#2 and was answered in answer 11 above.

REVIEWERS' COMMENTS:

Reviewer #1 (Remarks to the Author):

My comments are addressed and I recommend acceptance of the manuscript for publication in NC.

Reviewer #2 (Remarks to the Author):

This manuscript, showing the fabrication of arbitrary 3D structures using a sacrificial template replication, was revised by the authors, who also addressed all comments from the referees in detail. Specifically, the authors explained more in depth how their work differentiate with earlier work on femtolaser micromachining and fugitive inks. They also added experimental parameters and better detail the properties of the fabricated structures (surface chemistry/wettability, strength, roughness). An application based on hydrodynamic focusing was also demonstrated and the devices used for this were interfaced to standard tubings/pumping equipment. Consequently, I recommend publication of the revised manuscript as is.

Emmanuel Delamarche

Reviewer #3 (Remarks to the Author):

The author responded well to the reviewing comments. I recommend the paper to be published.